

# Variations in early life history traits of Japanese anchovy *Engraulis japonicus* in the Yangtze River Estuary

Chunlong Liu[1], Weiwei Xian[1,2], Shude Liu[1] and Yifeng Chen[3]

[1] CAS Key Laboratory of Marine Ecology and Environmental Sciences, Institute of Oceanology, Chinese Academy of Sciences, Qingdao, China
[2] Laboratory for Marine Ecology and Environmental Science, Qingdao National Laboratory for Marine Science and Technology, Qingdao, China
[3] Laboratory of Biological Invasion and Adaptive Evolution, Institute of Hydrobiology, Chinese Academy of Sciences, Wuhan, China

## ABSTRACT

Resources of Japanese anchovy (*Engraulis japonicus* Temminck & Schlegel, 1846) are undergoing dramatic recessions in China as the consequence of intensifying anthropogenic activities. Elucidating the influences of local-scale environmental factors on early life history traits is of great importance to design strategies conserving and restoring the declining anchovy resources. In this research, we studied hatching date and early growth of anchovy in the Yangtze River Estuary (YRE) using information obtained from otolith microstructure. Onset of hatching season and growth rates of anchovy was compared to populations in Japan and Taiwan. In YRE, the hatching date of anchovy ranged from February 26th to April 6th and mean growth rate ranged from 0.27 to 0.77 mm/d. Anchovies hatching later had higher growth rates than individuals hatching earlier before the 25th day. Among populations, hatching onsets of anchovy from the higher latitude were later than populations in the lower latitude, and growth rates of anchovy in YRE were much lower than populations in Japan and Taiwan. Variations in hatching onsets and early growth patterns of anchovy thus provide important knowledge on understanding the adaptation of anchovy in YRE and designing management strategies on conserving China's anchovy resources.

Corresponding author
Weiwei Xian, wwxian@qdio.ac.cn

## INTRODUCTION

Japanese anchovy (*Engraulis japonicus* Temminck & Schlegel, 1846) is a widespread fish in the northwest Pacific Ocean with great contributions to fishery resources and ecosystem functions (*Zhao et al., 2003*; *Takasuka & Aoki, 2006*; *Wan & Bian, 2012*). As the keystone species in coastal and marine ecosystems, anchovy plays the crucial role on connecting different trophic levels by acting as the predator for plankton and the prey for piscivorous fishes (*Iseki & Kiyomoto, 1997*; *Kim & Lo, 2001*; *Wang, Liu & Ye, 2006*; *Hsieh et al., 2009*). Due to the high susceptibility to environmental changes, anchovy resources exhibit remarkable fluctuations across years and regions (*Takahashi et al., 2001*; *Takasuka,*

*Oozeki & Aoki, 2007*). Elucidating the mechanisms responsible for fluctuations of anchovy resources is thus important to develop strategies for conserving coastal and marine biodiversity. Despite of the profound impacts of climatic factors on anchovy resources (*Kim & Lo, 2001*; *Hsieh et al., 2009*), local-scale environmental factors (e.g., sea surface temperature and habitat quality) are also the key determinants on dynamics of anchovy population, which has been well studied in Japan, Korea and Taiwan (*Chen & Chiu, 2003*; *Takahashi & Watanabe, 2004*; *Takasuka, Oozeki & Aoki, 2007*). However in mainland China, the region owning the widest distributions and largest catches of anchovy (Fisheries and Aquaculture Department, FAO, Rome, Italy; available: http://www.fao.org/home/en/), little effort was paid to explore the influence of local environmental factors on anchovy populations (but see *Zhu, Zhao & Li, 2007*).

Early life history in larval stage is the "window" in which fish has the highest vulnerability and mortality rates (*Takasuka, Aoki & Mitani, 2003*; *Takahashi & Watanabe, 2004*; *Starrs, Ebner & Fulton, 2016*). In early life history, growth rate is the main factor determining larval duration and mortality, which are both closely related to recruitment strength (*Takasuka, Aoki & Mitani, 2003*; *Takahashi & Watanabe, 2004*; *Starrs, Ebner & Fulton, 2016*). Within a population, fish with faster growth could gain the larger size by accelerating the development and metamorphosis compared to other individuals from the same cohort, consequently having the shorter larval duration and higher survival rate ("growth-mortality" hypothesis; *Takahashi et al., 2001*; *Hwang et al., 2006*; *Takasuka & Aoki, 2006*; *Itoh et al., 2011*). Early growth of anchovy is significantly affected by local environmental factors and even slight environmental changes might cause great variations in population mortality and the amount of recruitments into the adult population (*Takahashi et al., 2001*; *Chen & Chiu, 2003*; *Takahashi & Watanabe, 2004*; *Hwang et al., 2006*; *Zenitani et al., 2009*). Water temperature is one of the most important factors affecting anchovy early growth (*Hwang et al., 2006*; *Takasuka, Oozeki & Aoki, 2007*). When temperature is lower than the optimal growth temperature of fish, higher temperature would consistently accelerate fish growth by improving individual metabolic rate (*Hwang et al., 2006*; *Takasuka, Oozeki & Aoki, 2007*). In addition, a number of studies have emphasized the importance of habitat quality on fish growth (e.g., *Amara et al., 2007*; *Amara et al., 2009*). The degradation in habitat quality decelerates fish growth through reducing food availability and directing fish to devote more energy for tolerating higher pollution (*Amara et al., 2007*; *Amara et al., 2009*). Understanding variations in early growth of anchovy under different environments could therefore provide crucial insights on predicting dynamics and estimating the year-class recruitment strength of anchovy population (*Takasuka, Oozeki & Aoki, 2007*).

China's anchovy resources are undergoing dramatic declines as the consequence of dam construction, intensified overfishing and water pollution (*Zhao et al., 2003*; *Zhu, Zhao & Li, 2007*). The recession in anchovy resources is especially striking between 1993 and 2002, with annual catch rapidly decreasing from 4.12 to 0.18 million tons and posing significant threats to fishery economy and ecosystem functions (*Zhao et al., 2003*). To conserve and restore the declining anchovy resources, it's urgently needed to design applicable management strategies to ensure the success of population recruitment based on the adaptivity of anchovy to China's environments (*Hwang et al., 2006*; *Wang, Liu & Ye,*

*2006*). In consideration of the key role of growth in the process of population recruiting, determining the impacts of environmental changes on early growth could shed important lights on elucidating anchovy adaptivity in the early life history (*Takasuka & Aoki, 2006*; *Takasuka, Oozeki & Aoki, 2007*).

The Yangtze River Estuary (YRE) is an important spawning, feeding and nursery ground for Japanese anchovy and other fishes benefiting from the high productivity contributed by abundant sediments in the outflow of the Yangtze River (*Zhou, Shen & Yu, 2008*; *Yu & Xian, 2009*; *Zhang et al., 2009*). However, intensifying urbanization and increasing anthropogenic activities are causing severe degradations of the aquatic ecosystem in the YRE, leading to remarkable declines in anchovy resources (*Jiao et al., 2007*). To provide knowledge on conserving anchovy resources in YRE, we aim for achieving two overarching targets using information obtained from otolith microstructure: (1) to determine the growth pattern in early life history of anchovy from YRE; (2) to detect variations in anchovy early growth among groups with different hatching dates and among populations across the northwestern Pacific Ocean. By revealing the intra- and inter-population differences in early growth patterns, our study could contribute valuable information to the development of cost-effective strategies on managing anchovy resources in the highly exploited aquatic ecosystems of YRE.

## MATERIALS AND METHODS

### Field sampling

Anchovy larvae were collected in May, 2012 in the "Spring investigation of fishery resources and ecology in Yangtze River Estuary" survey. Forty stations were set from the mouth of the Yangtze River to the offshore (30°45′N–30°45′N, 122°20′E–123°20′N) (*Xing, Xian & Shen, 2014*; *Li et al., 2015*). In this survey, environmental factors showed noticeable variations among stations, indicating the very high physical and chemical heterogeneity in YRE. For example, the depth ranged from 3 to 60 m and the salinity ranged from 0.13 to 33.98‰ (Supplemental Information 1). At each station, a horizontal plankton net (0.8 m diameter with 0.5 mm mesh size) was towed at the surface with a speed of two knots for ten minutes to sample anchovy. During this survey, anchovy were collected in two stations (Station 29, 1,129 individuals; Station 30, 1,342 individuals). Sampled larvae were immediately preserved in 90% ethanol and taken back to the laboratory. All specimens were collected in accordance with wild animal conservation law issued by the People's Republic of China for the purposes of conducting research on Japanese anchovy.

### Environmental data

Data of daily sea surface temperatures (SST) in each station were obtained from NOAA SST High Resolution Dataset (http://www.esrl.noaa.gov/psd/) to represent the water temperature across anchovy growing seasons (from February 26th to May 3rd, 2012; *see* 'Results'). Daily SST data are generated from an Advanced Very High Resolution Radiometer (AVHRR), which can infer the precise SST at very high resolution (1.09 km) using multi spectral analysis. The time series SST data allowed us to assess the influence

of water temperature on anchovy growth by comparing growth patterns among individual hatching on different dates.

## Otolith microstructure analysis

A subset of 200 individuals were randomly selected from samples at Station 29 and 30. Standard length (SL) of each individual was measured to the nearest mm, and both right and left sagittal otoliths were extracted from fish head under a dissecting microscope. Either the right or left otolith was mounted on a slide using melting thermoplastic glue and polished with 15 µm lapping film until increments could be clearly interpreted (*Wang & Tzeng, 1999*). Each unbroken otolith section was photographed at 400× magnification using a digital camera fixed to a light microscope (BH2, Olympus Optical Co. Ltd., Tokyo, Japan) to obtain the picture of each otolith section with clear increments. Numbers and widths of otolith increments were counted and measured along the maximum otolith radius (OR) from the nucleus to the edge using Increment Analysis Program (Huazhong Agricultural University, Wuhan, Hubei Province, China). For each section, we made two independent measurements on increment numbers. If the difference in two numbers differed less than 5%, one number was randomly selected as the increment number of this otolith; otherwise the increment number was measured again. If the third number differed by <5% compared to one of the first two numbers, the third number was used as the increment number. If the third evaluation still differed the first two numbers by >5%, that otolith section was discarded.

## Data analysis

The daily age (D) of each individual was determined using the increment number plus three, because the first increment of anchovy otolith is deposited on the fourth day after hatching (*Tsuji & Aoyama, 1984*). Hatch dates were thus back-calculated by subtracting age from the catch date (May 3rd). Daily somatic growth rates were back-calculated from increment width using the biological intercept method (*Campana, 1990*), with the length at hatching (5.6 mm) as the biological intercept (*Tsuji & Aoyama, 1984*). Data normality test indicated that the variances of SL and D were not equal for anchovy from two stations, and Wilcoxon signed-rank tests were thus used to compare frequency distributions of both SL and D between stations. Linear regression was used to fit relationships between SL and OR and between SL and D. Non-parametric repeated measures analysis of covariance was performed to compare the relationship of SL with D between two stations. Because of the significantly positive relationships between OR and SL (see 'Result'), otolith increment widths were used as the proxy of anchovy early growth rates.

Given the possible differences in early growth among individuals hatching on different dates, anchovy from each station were divided into three nearly equal-sized groups according to hatching dates to compare otolith growth trajectories of anchovy within population. Anchovy hatching from February 26th to March 16th were categorized "early group", from March 17th to March 26th as "middle group", and from March 27th to April 6th as "late group". A repeated measures analysis of variance (RM-ANOVA) was used to compare otolith growth trajectories among groups in each station (*Searcy & Sponaugle,*

*2000*). Because the minimum age of anchovy was 24 days, the level of RM-ANOVA was set at 24 to include all samples. The within-subject factor was daily growth rate and the between-subject factor was group. Because the distribution of increment widths was not normal, data of increment widths were log-transformed before RM-ANOVA.

Compared to anchovy in YRE, populations in Japan and Taiwan should be less affected by pollutions and overfishing due to the better conservation on coastal and marine environments (*Kim & Lo, 2001*; *Chai et al., 2006*; *Wang, Liu & Ye, 2006*; *Takasuka, Oozeki & Aoki, 2007*). To assess the difference in early growth patterns of anchovy living under different habitat qualities, mean growth rate of each anchovy were calculated for the comparison in growth of anchovy in Japan and Taiwan. To do so, we conducted the extensive searches on data of hatching onset and early growth rates of anchovy from scientific papers. Only studies reporting both two characters were kept. For each region, mean growth rates of anchovy in populations with earliest hatching dates were selected as the representative to compare growth of the first emerging individuals anchovy in each region. Consequently, data of anchovy in Taiwan (*Chiu & Chen, 2001*) and Japan (*Takahashi et al., 2001*) were used for inter-population comparisons. All the analyses were performed in R 3.2.0 (*R Development Core Team, 2014*) using the packages *sm* (*Bowman & Azzalini, 2014*).

## RESULTS

There was a continuous increase in SST during anchovy growth season (Fig. 1; Supplemental Information 2). SST rose from 6.2 to 16.6 °C at Station 29 and from 6.7 to 16.0 °C at Station 30 between 29th February and 3rd May. The consistent rising in SST indicated that the early, middle and late groups experienced different thermal environments during their growing seasons.

SL and D of anchovy at Station 29 were both significantly lower than Station 30 (Fig. 2; Table 1), demonstrating anchovy were larger and older in the offshore. Hatch dates ranged from February 26th to April 6th at Station 29, and from February 28th to April 5th at Station 30 (Fig. 2; Table 1). Growth rates of anchovy ranged from 0.27 to 0.77 mm/d and from 0.29 to 0.73 mm/d at Station 29 and 30 respectively (Table 1; Supplemental Information 3). Relationships between SL and OR were both significantly positive at two stations (both $p < 0.01$). No significant difference was found in the relationships of SL with D between two stations ($p > 0.05$), and the SL-D relationship were described by a common regression equation for two populations: $SL = 0.386D + 4.87$ ($R^2 = 0.55$, $p < 0.01$). Despite of the overall similar patterns of increasing growth rates for three groups before the 25th day, significant differences were found in growth trajectories between three groups at each station (Station 29, all $p < 0.05$; Station 30, all $p < 0.05$), with anchovy hatching later showing much faster growth (Fig. 3). After the 25th day, differences in growth trajectory among groups became larger, with growth rates being highest for early groups and lowest for late groups.

The ranges of standard length were largely overlapped for three populations, suggesting the similar ontogenic stage of anchovy from Japan, YRE and Taiwan (Table 2). There was a

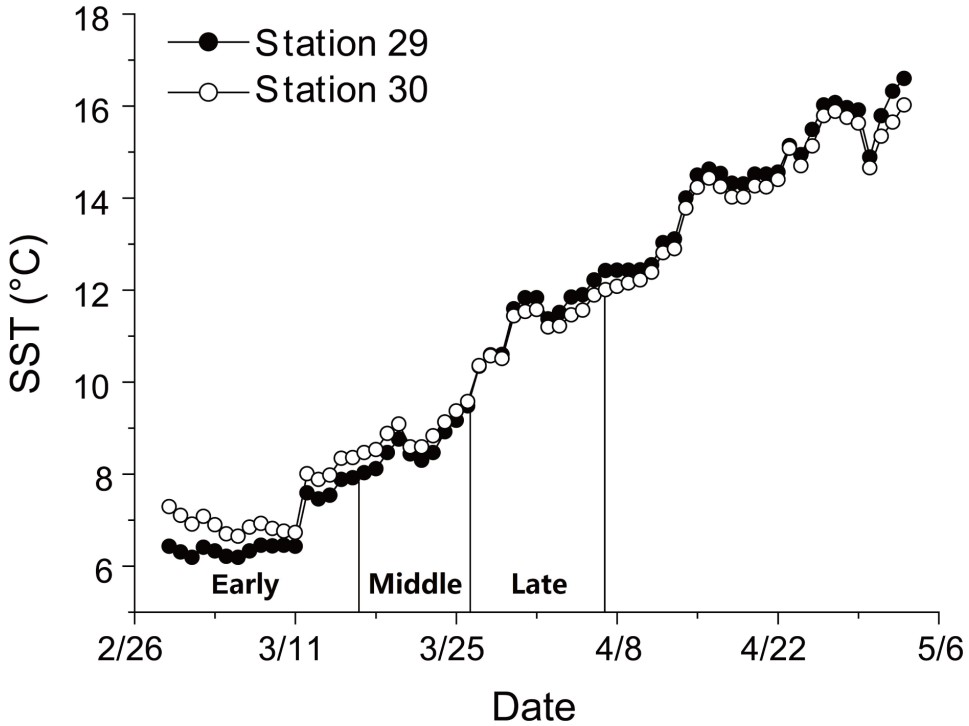

**Figure 1** **Changes in sea surface temperature (SST).** Trends in sea surface temperature (SST) at Stations 29 and 30 during growing season of anchovy in the Yangtze River Estuary. Anchovy in each station were divided into three groups (Early, Middle and Late) based on their hatch dates, which were shown on the *X* axis.

counter­gradient pattern in hatching onsets among populations, with the onset being later with increasing latitude in a rate of 15 d per five degrees. However, no consistent pattern was found between growth rates and latitude. The minimum and maximum growth rates of anchovy from YRE were both clearly lower than those of anchovy from Taiwan and Japan. The highest growth rate of YRE population was even lower than the minimum growth rate of population in Japan.

## DISCUSSION

Understanding the response of fish early growth to environmental changes is of great importance to predict the strength of population recruitment and dynamics of fishery resources (*Takasuka, Aoki & Mitani, 2003*; *Takasuka & Aoki, 2006*). Comparing growth rates and growth patterns of individuals from different populations or groups is a common way to investigate the influences of environmental changes on fish early growth (*Chen & Chiu, 2003*; *Yasue & Takasuka, 2009*). In this study, we found significant differences in early growth of anchovy hatching on different dates and among populations from different regions. Variations in growth patterns on the temporal and spatial scales provide important knowledge on elucidating the adaptation of anchovy in YRE and facilitating the conservation of anchovy resources across the northwestern Pacific Ocean.

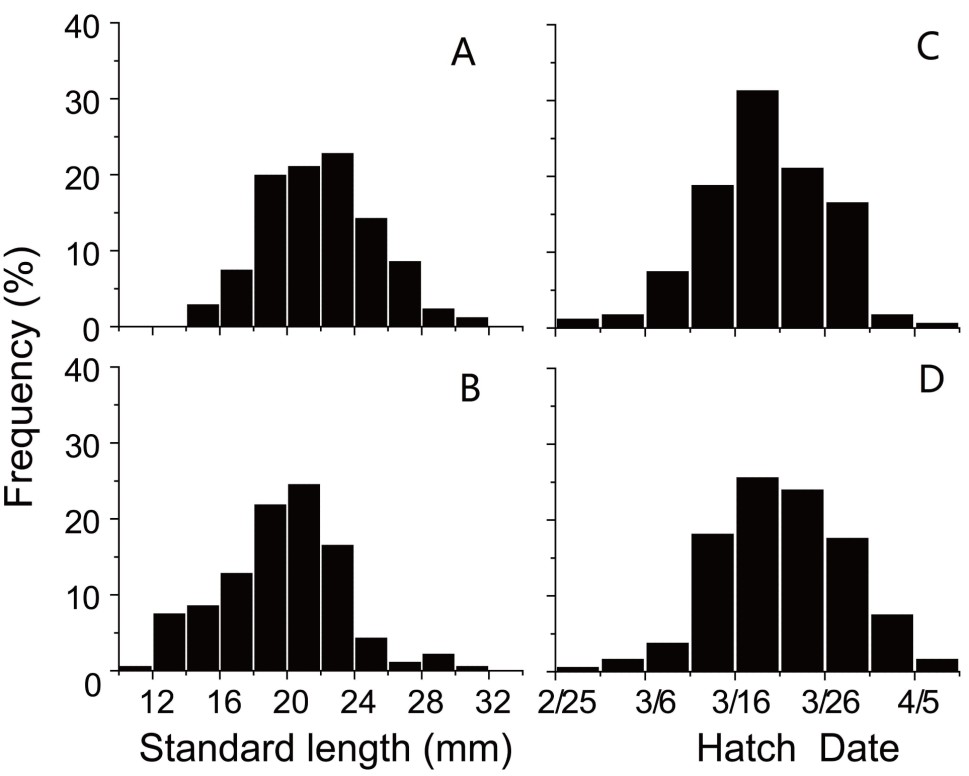

**Figure 2** **Distributions of standard length and hatching date.** Frequency distributions of (A) standard lengths of anchovy at Station 29; (B) standard lengths of anchovy at Station 30; (C) hatching dates of anchovy at Station 29; (D) hatching dates of anchovy at Station 30.

**Table 1** **Biological data of anchovy.** Number ($N$), standard length, hatching date and mean growth rate of anchovy from three groups at the Stations 29 and 30 in the Yangtze River Estuary. Three groups were divided according to individual hatching dates. Hatching date and growth rate were back-calculated from the number and width of increments deposited in otolith section.

| Station | Group | N | Standard length (mm) | | Hatching date | Mean growth rate (mm d$^{-1}$) | |
|---|---|---|---|---|---|---|---|
| | | | Mean | Range | | Mean | Range |
| 29 | Early | 51 | 22.67 | 16.72–32.00 | 2/26–3/16 | 0.38 | 0.27–0.55 |
| 29 | Middle | 96 | 19.63 | 12.93–29.27 | 3/17–3/26 | 0.46 | 0.30–0.59 |
| 29 | Late | 41 | 15.90 | 11.07–20.87 | 3/27–4/6 | 0.51 | 0.37–0.77 |
| 30 | Early | 55 | 24.59 | 18.81–30.12 | 2/28–3/16 | 0.41 | 0.29–0.56 |
| 30 | Middle | 94 | 21.47 | 14.62–28.63 | 3/17–3/26 | 0.50 | 0.38–0.68 |
| 30 | Late | 27 | 17.93 | 14.02–21.87 | 3/30–4/5 | 0.55 | 0.42–0.73 |

Hatching date is proven as the key factor affecting anchovy growth during the early life history, with growth rates showing an increasing pattern with hatching date. Differences in early growth of anchovy hatching on different dates have been reported for populations in the Yellow Sea (*Hwang et al., 2006*) and the East China Sea (*Takasuka & Aoki, 2006*). Increasing temperature is attributed as the primary factor accelerating anchovy early growth. Growth rates of anchovy are found positively related to temperature up to 20–26 °C

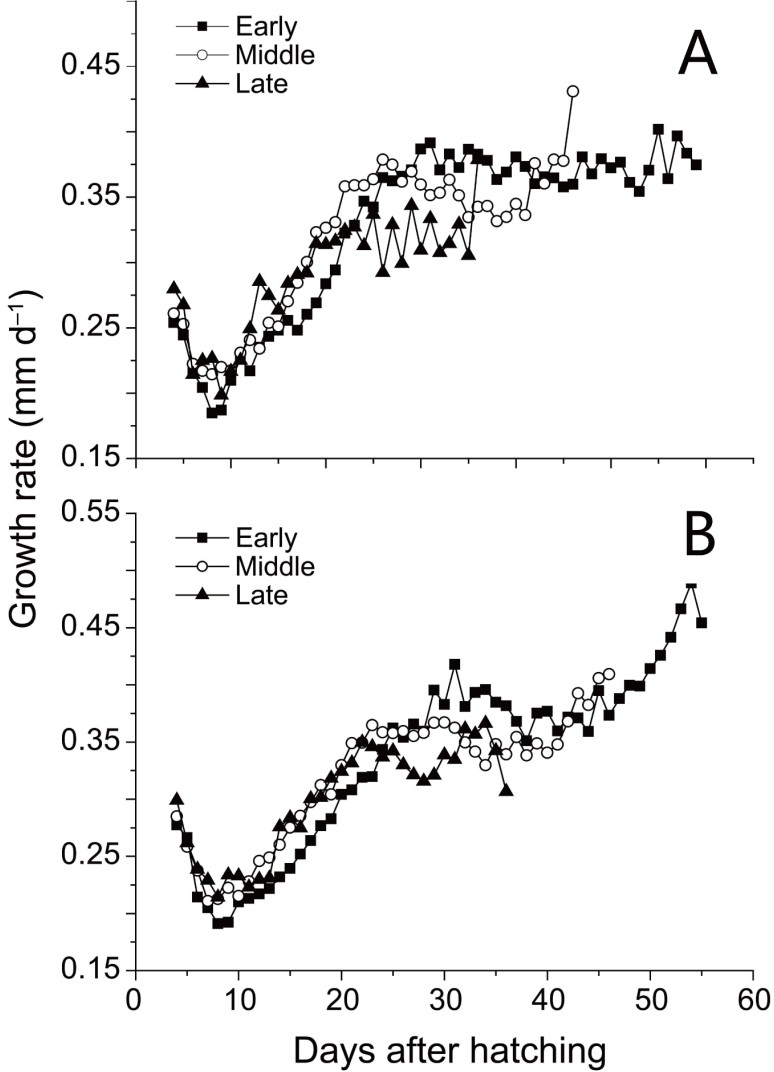

**Figure 3  Growth trajectories of anchovy.** Mean growth trajectories of anchovy from three groups at Stations 29 and 30 (excluding days where anchovy numbers were less than five). Growth rates were back-calculated from increment widths in otolith microstructure using the biological intercept method.

(*Hwang et al., 2006*; *Takasuka & Aoki, 2006*). During the growth season beginning from February, SST in YRE rapidly rises from 6 °C to 16 °C in May, thus probably increasing growth rates by improving individual metabolic rate and prey abundance. However, higher growth rates in the beginning may not compensate for the shorter growing season of late-hatching anchovy because of their lower growth rates compared to early-hatching anchovy after the 25th day.

It is important to note that growth rates of anchovy in YRE are much lower than populations in Japan and Taiwan. This result is unexpected as the high productivity in YRE should have supported the faster early growth of anchovy (*Zhou, Shen & Yu, 2008*). Although there are difficulties in determining the specific mechanisms, several factors

**Table 2  Information on three anchovy populations.** Distribution, standard length, hatching onset and mean growth rate of anchovy in populations from Taiwan, the Yangtze River Estuary and Japan.

| Region | Latitude | Standard length (mm) | | Hatching onset | Mean growth rate (mm d⁻¹) | | Source |
|---|---|---|---|---|---|---|---|
| | | Minimum | Maximum | | Minimum | Maximum | |
| Taiwan | 24–25°N | 17.2 | 31.3 | 2/12 | 0.37 | 0.91 | *Chiu & Chen (2001)* |
| The Yangtze River Estuary | 30–31°N | 11.07 | 32 | 2/26 | 0.2 | 0.46 | Present study |
| Japan | 35–40°N | 20 | 35 | 3/3 | 0.49 | 0.71 | *Takahashi et al. (2001)* |

might be responsible for the lower growth rates. First, polluted waters could decelerate fish growth by directing more energy toward the tolerance of worse conditions (*Amara et al., 2007*; *Amara et al., 2009*). Aquatic ecosystems of YRE are threatened by increasing anthropogenic activities and sewage discharges with an unparalleled magnitude (*Wang, Liu & Ye, 2006*; *Jiao et al., 2007*). To tolerate the degrading environments, anchovy have to reduce the energy devoted for growth and consequently have slower growth. Second, the mis-match of hatching season with the occurrence of optimal conditions might be another factor decelerating fish growth ("match/mismatch hypothesis"; *Frank & Leggett, 1982*; *Takasuka, Aoki & Mitani, 2003*; *Takahashi & Watanabe, 2004*). Due to the influence of the Asian monsoon on runoff in the Yangtze River, water and sediment flowing into in YRE have clear monthly changes (*Jiang et al., 2014*; *Tang, Li & Chen, 2018*). In February, water and sediment reach to the lowest values across the year, decreasing the nutrition supporting the growth of plankton. Anchovy hatching from February therefore have the lower growth rate due to lower temperature and insufficient food. Third, variations of intrinsic attributes among populations would also contribute to the difference in early growth. Growth rates of fish are determined by the interplay of phenotypic plasticity and genetic adaptivity (*Conover & Present, 1990*; *Sexton, McKay & Sala, 2002*; *Liu et al., 2015*). Across the northwestern Pacific Ocean, there might be substantial differences in phenotypic and genetic attributes among populations in Japan, YRE and Taiwan, causing unregular spatial patterns in growth rates. Overall, the lower growth rates will contribute to smaller length of anchovy in YRE by the end of the first growth season, finally decreasing overwinter survival rates and the strength of population recruitment (*Amara et al., 2007*; *Amara et al., 2009*). Future strategies on conserving anchovy resources should take the inter-population variations in early growth into account to accelerate population recruitment.

Later hatching onset of anchovy in higher latitude reveals their shorter first growing season, which is in accordance with the pattern of "counter-gradient variation". The decrease in length of growing season is applicable for other marine fishes, such as *Menidia menidia* (*Conover & Present, 1990*) and *Morone saxatilis* (*Conover, Brown & Ehtisham, 1997*). Changes in environmental factors might be the main driver for variations of hatching onset. In the higher latitude, lower water temperature and shorter daytime contribute to the lower metabolic rate and growth rates, impeding the maturation and reproduction of fish (*Conover & Present, 1990*; *Tarkan, 2006*; *Benejam et al., 2009*; *Carmona-Catot, Benito & García-Berthou, 2011*). Additionally, lower temperature decelerates gonad development by depressing the growth of plankton and copepods (*Hwang et al., 2006*; *Tanaka et al., 2008*).

Delayed maturation and spawning therefore result in the later fish hatching in higher latitude. Given the wide distribution of anchovy across the northwestern Pacific Ocean, variations in hatching onset among populations should be integrated into management strategies of anchovy resources to better back-calculate the spawning season and predict dynamics of population recruitment.

Intensifying anthropogenic activities and environmental pollutions are threatening the function of estuarine ecosystems (*Gilliers et al., 2006*; *Amara et al., 2009*; *Bacheler et al., 2009*). Elucidating the influence of environmental changes on fish early life history traits is crucial for estimating year-class strength of population recruitment and annual fishery resources (*Wang & Tzeng, 1999*; *Takasuka, Oozeki & Aoki, 2007*). Our results suggest that shifts in hatching dates and growth patterns play a key role on anchovy adaptivity in YRE and across the northwestern Pacific Ocean. In consideration of the increasing impacts of climate changes and habitat modifications on estuarine ecosystems (*Zhou, Shen & Yu, 2008*; *Zhang et al., 2009*), future researches should quantify the influence of different environmental factors on anchovy early growth to provide useful information on restoring anchovy resources and conserving fish diversity in YRE and other estuaries.

# ACKNOWLEDGEMENTS

We thank Yushun Chen and three anonymous reviewers for their helpful comments on the manuscript. We thank Liwen Bianji, Edanz Editing China (http://www.liwenbianji.cn/ac), for the editing of an English text of a draft of this manuscript.

## Funding

This research is supported by the NSFC-Shandong Joint Fund for Marine Ecology and Environmental Sciences (U1606404), National Natural Science Foundation of China (No. 31272663, No. 41176138) and Qingdao Post-doctor Application Research Project (Y8KY01106N). The funders had no role in study design, data collection and analysis, decision to publish, or preparation of the manuscript.

## Grant Disclosures

The following grant information was disclosed by the authors:
NSFC-Shandong Joint Fund for Marine Ecology and Environmental Sciences: U1606404.
National Natural Science Foundation of China: 31272663, 41176138.
Qingdao Post-doctor Application Research Project: Y8KY01106N.

## Competing Interests

The authors declare there are no competing interests.

## Author Contributions

- Chunlong Liu conceived and designed the experiments, performed the experiments, analyzed the data, contributed reagents/materials/analysis tools, prepared figures and/or tables, authored or reviewed drafts of the paper.

- Weiwei Xian conceived and designed the experiments, performed the experiments, prepared figures and/or tables, authored or reviewed drafts of the paper.
- Shude Liu conceived and designed the experiments, performed the experiments.
- Yifeng Chen conceived and designed the experiments, authored or reviewed drafts of the paper.

## Field Study Permissions

The following information was supplied relating to field study approvals (i.e., approving body and any reference numbers):

The field permit was provided by the key station of the monitoring system on ecology and environment of the Three Gorge Dam: 2013-011; JC-011606.

## Data Availability

The raw data are provided in the Supplemental Files.

## Supplemental Information

Supplemental information for this article can be found online at http://dx.doi.org/10.7717/peerj.4789#supplemental-information.

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
