# Peer review of "Variations in early life history traits of Japanese anchovy Engraulis japonicus in the Yangtze River Estuary"

_PeerJ, doi:10.7717/peerj.4789_

## Round 0.1 · original submission · Major Revisions

I concur with the three expert reviewers that the manuscript requires a Major Revision. When submitting a revised version, please provide a list of point-by-point responses to individual comments of the reviewers.

·

Basic reporting

There are some minor issues with presentation of the text (i.e. spaces between words are occasionally omitted throughout) but the use of English language is quite good.

The use of references from the literature could use some improvement, as there are huge amounts of literature available on anchovy and other similar fish from many regions regarding their growth and analysis of otoliths, but only a small amount of this literature is assessed in this manuscript. Also, some of the references do not seem to be on the topics of the sentences that are placed with e.g. line 36-37. And some large statements are made without providing references, which should be included e.g. line 43.

The figures are good quality, but some of them are not very relevant and are likely not needed. For example, Figure 1 shows a large number of sampling stations, but it appears only 2 of these stations were actually used here, so most of the information is unnecessary. Figure 2 only serves the purpose of showing a photograph of an otolith, which does not provide much information to the reader. Figure 5 and 6 show that as anchovy get older in age, they also get bigger in length and their otoliths get bigger, which is only very basic information which probably does not need to be presented.

Experimental design

The research aims to fill a knowledge gap concerning unknown aspects of early life history of anchovy in the Yangtze estuary, but the design of the experiment is small in scale and in its current form cannot come close to achieving this aim. It appears only two closely spaced stations were assessed here with samples taken during 10 minutes of sampling at each station. Readers will wonder why anchovy samples were not also collected and analysed from the other 32 stations. For example, an analysis of effects from a possible salinity gradient from the stations within the river to the many off-shore stations may make the study more comprehensive and of greater interest. The authors should consider possibilities like this for any re-writing of the manuscript.

From the abstract and introduction it seems to the reader that some new data will be presented also about anchovy in Taiwan and Japan, but when the reader gets to the methods it is clear these data are previously published descriptions from many years previous that the Yangtze data are very informally compared to. Thus the aim stated of inter-population comparisons is also not achieved.

The quantification of otolith properties from the two stations and calculation of population data appears to be done to a high technical standard, but the statistical analysis needs improvement, e.g. the assumptions (data normality and variance homogeneity) need to be tested and more details of the procedures need to be provided, such as the statistical programmes used for analyses.

Validity of the findings

As stated above, more comprehensive data will need to be added to this study before it can be valid for providing relevant information on anchovy in this region. If the authors wish for some valid comparisons with other regions such as Taiwan and Japan, actual new data from those regions will be to be analysed alongside the data from the Yangtze.

Reviewer 2 ·

Basic reporting

The language and formatting need some improvements. Many words are fused as a single word and some sentences e.g. line no 37, 67-69, 91-92 can be simplified for clarity.

Experimental design

Authors aim to profile the distribution of Engraulis japonicus and their larvae to understand variations in their growth and life history traits. A few suggestions are listed below:

- It is not very clear when is adult fish data discussed and when is the larvae data discussed. This is confusing in both text and figures. It will be good to explicitly indicate larvae and adult related results for clarity in both text and figures. For example, is otolith analyzed from larvae, fish or both? How is the growth rate of the fish calculated.

- Line 118-120: cite the source of the data for other regions.

- The data on all the individual larvae should be provided as supplemental files. Also, the annotation of the supplemental file is not enough to understand the columns.

- Authors should also use a non-parametric method to compare frequency distributions.

- It will be great to include pictures of larvae and the fish if possible.

Validity of the findings

The manuscript is mostly descriptive and the scope may be very limited. The findings may be valuable for researchers or conservation biologists working on this specific fish. It will be great to discuss the implications of their findings for the conservation of this fish.

Reviewer 3 ·

Basic reporting

The paper was generally written clearly, but both the introduction and discussion could be much more thorough. Please see my attached review.

Experimental design

The experimental design was appropriate and the study was carried out properly. Methods used were well established and appropriate. I only have issue with the use of the biological intercept method in an area and on a population where otolith and somatic growth may be de-coupled. I discuss this more in depth in the attached review.

Validity of the findings

Here is where I think the manuscript needs further development. I don't feel the authors have thoroughly discussed the patterns they have identified. All the possible drivers of these patterns should be addressed and they should argue their hypotheses of why they found what they did.

Annotated reviews are not available for download in order to protect the identity of reviewers who chose to remain anonymous.

---

## Round 0.2 · accepted · Accept

I have read the rebuttals and revised MS which are in good order. The study is of good quality and thus I am very pleased to Accept this revised manuscript for publication.

Reviewer 2 ·

Basic reporting

The manuscript, particularly the introduction is much improved.

Experimental design

No comments.

Validity of the findings

The findings will still be important for only Anchovy researchers, but given the importance of the fish for the region, they can be considered significant.

Additional comments

The authors have responded to my comments to satisfaction, and I have no further questions.